# Sequencing Treatments in Patients with Advanced Well-Differentiated Pancreatic Neuroendocrine Tumor (pNET): Results from a Large Multicenter Italian Cohort

**DOI:** 10.3390/jcm13072074

**Published:** 2024-04-03

**Authors:** Francesco Panzuto, Elisa Andrini, Giuseppe Lamberti, Sara Pusceddu, Maria Rinzivillo, Fabio Gelsomino, Alessandra Raimondi, Alberto Bongiovanni, Maria Vittoria Davì, Mauro Cives, Maria Pia Brizzi, Irene Persano, Maria Chiara Zatelli, Ivana Puliafito, Salvatore Tafuto, Davide Campana

**Affiliations:** 1Digestive Disease Unit, Sant’Andrea University Hospital, ENETS Center of Excellence, 00189 Rome, Italy; francesco.panzuto@uniroma1.it (F.P.); mrinzivillo@ospedalesantandrea.it (M.R.); 2Department of Medical-Surgical Sciences and Translational Medicine, Sapienza University of Roma, 00189 Roma, Italy; 3Department of Medical or Surgical Sciences, University of Bologna, 40126 Bologna, Italy; elisa.andrini3@unibo.it (E.A.); davide.campana@unibo.it (D.C.); 4Division of Medical Oncology, IRCCS Azienda Ospedaliera–Universitaria Bologna, Neuroendocrine Tumor Team Bologna, ENETS Center of Excellence Bologna, 40138 Bologna, Italy; 5Department of Medical Oncology, Fondazione IRCCS Istituto Nazionale dei Tumori di Milano, European Neuroendocrine Tumor Society (ENETS) Center of Excellence, 20133 Milan, Italy; sara.pusceddu@istitutotumori.mi.it (S.P.); alessandra.raimondi@istitutotumori.mi.it (A.R.); 6Department of Oncology and Hematology, Division of Oncology, University Hospital of Modena, 41121 Modena, Italy; fabiogelsomino83@yahoo.it; 7Osteoncology and Rare Tumors Center, IRCCS Istituto Romagnolo per lo Studio dei Tumori “Dino Amadori”, 47014 Meldola, Italy; alberto.bongiovanni@irst.emr.it; 8Department of Medicine, Section of Endocrinology, University and Hospital Trust of Verona, ENETS Center of Excellence, 37129 Verona, Italy; mariavittoria.davi@aovr.veneto.it; 9Department of Interdisciplinary Medicine, University of Bari, 70121 Bari, Italy; mauro.cives@uniba.it; 10Division of Medical Oncology, A.O.U. Consorziale Policlinico di Bari, 70124 Bari, Italy; 11Division of Medical Oncology, Azienda Ospedaliera Universitaria San Luigi Gonzaga, 10143 Orbassano, Italy; brizzimariapia@gmail.com (M.P.B.); irene.persano@unito.it (I.P.); 12Department of Medical Sciences, Section of Endocrinology, Geriatrics and Internal Medicine, University of Ferrara, 44121 Ferrara, Italy; ztlmch@unife.it; 13Oncologia Medica, Istituto Oncologico del Mediterraneo, 95029 Viagrande, Italy; ivana.puliafito@grupposamed.com; 14Oncologia Clinica e Sperimentale Sarcomi e Tumori Rari, Istituto Nazionale Tumori IRCCS, Fondazione G. Pascale, 80131 Naples, Italy; s.tafuto@istitutotumori.na.it

**Keywords:** pancreatic NET, SSA, RLT, targeted therapy, GEP-NENs, treatment sequence

## Abstract

**Background:** The optimal treatment sequencing for advanced, well-differentiated pancreatic neuroendocrine tumors (pNETs) is unknown. We performed a multicenter, retrospective study to evaluate the best treatment sequence in terms of progression-free survival to first-line (PFS1) and to second-line (PFS2), and overall survival among patients with advanced, well-differentiated pNETs. **Methods**: This multicenter study retrospectively analyzed the prospectively collected data of patients with sporadic well-differentiated pNETs who received at least two consecutive therapeutic lines, with evidence of radiological disease progression before change of treatment lines. **Results**: Among 201 patients, 40 (19.9%) had a grade 1 and 149 (74.1%) a grade 2 pNET. Primary tumor resection was performed in 98 patients (48.8%). First-line therapy was performed in 128 patients with somatostatin analogs (SSA), 35 received SSA + radioligand therapy (RLT), 21 temozolomide-based chemotherapy, and 17 SSA + targeted therapy. PFS was significantly longer in patients with grade 1 pNETs compared to those with grade 2, in patients who received primary tumor surgery, and in patients treated with RLT compared to other treatments. At multivariate analysis, the use of upfront RLT was independently associated with improved PFS compared to SSA. Second-line therapy was performed in 94 patients with SSA + targeted therapy, 35 received chemotherapy, 45 SSA + RLT, and 27 nonconventional-dose SSA or SSA switch. PFS was significantly longer in patients treated with RLT compared to other treatments. At multivariate analysis, the type of second-line therapy was independently associated with the risk for progression. OS was significantly longer in patients who received primary tumor surgery, with Ki67 < 10%, without extrahepatic disease, and in patients who received SSA–RLT sequence compared to other sequences. **Conclusions**: In this large, multicenter study, RLT was associated with better PFS compared to other treatments, and the SSA–RLT sequence was associated with the best survival outcomes in patients with pNETs with Ki67 < 10%. Primary tumor surgery was also associated with improved survival.

## 1. Introduction

Neuroendocrine neoplasms (NENs) are a heterogeneous group of tumors, characterized by a variable range of differentiation, grading, primary sites, and functional status. According to the 2019 World Health Organization (WHO), NENs are classified into well-differentiated tumors, with grading G1, G2, or G3 based on Ki67 value (<3%, ≥3–20%, and >20%, respectively), and poorly-differentiated neuroendocrine carcinomas (NECs) [1]. Over the past few decades, NEN incidence has increased, with an age-adjusted incidence rate of 6.4-fold increase from 1973 to 2012, and prevalence increase from 0.006% in 1993 to 0.048% in 2012, including all tumor sites, stages, and grades [2]. The increased incidence is primarily due to improved detection of early-stage disease, whereas increased prevalence is linked to improvement of treatment strategies and survival outcomes, but also to improvement of diagnostic techniques detecting more incidental tumors and to a greater awareness among pathologists [3]. Historically, NENs are considered tumors with an indolent behavior and favorable long-term outcomes [4]. However, the prognosis is widely variable based on several factors, including tumor histology, grading, primary site of tumor, and stage at diagnosis. Currently available treatment strategies in well-differentiated advanced NETs include somatostatin analogs (SSA), targeted therapies (i.e., everolimus and sunitinib), chemotherapy, such as temozolomide-based treatment, and radioligand therapy (RLT) with lutetium 177 (^177^Lu)-DOTATATE. Because of the lack of head-to-head comparison studies, the optimal treatment sequencing has not yet been defined. According to clinical practice, the sequencing of systemic treatments is still individualized and based on tumor type, grading, somatostatin receptor (SSTR) expression, extent of disease, and patient-related characteristics (e.g., age, performance status, comorbidities, presence of symptoms, and patient preferences). Therefore, the clinical decision-making and management of NENs is complex and requires discussion within a multidisciplinary team, which should include medical and radiation oncologists, surgeons, radiologists, and nuclear medicine specialists, at high-volume referral centers. This is particularly important for patients with advanced, well-differentiated pancreatic neuroendocrine tumors (pNETs), considering that the NETTER-1 trial proved the safety and efficacy of RLT as a second-line therapy after progression on standard SSA only in patients with well-differentiated midgut NENs [5]. Recently, preliminary results from the phase II randomized OCLURANDOM trial of RLT vs. sunitinib in patients with pNETs have been presented, showing an almost double progression-free survival rate at 12 months, the primary endpoint, but the final report is still awaited [6]. Thus, prospective data on the efficacy of RLT compared to chemotherapy or targeted therapy in patients with pNETs are still lacking. The identification of the best treatment timing and sequence in patients with advanced well-differentiated pNETs represents a critical unmet clinical need. Here, we performed a multicenter, retrospective study involving 11 Italian NET referral centers to evaluate the optimal treatment sequencing in a large population of patients with advanced well-differentiated pNETs.

## 2. Materials and Methods

### 2.1. Study Design

The study included all consecutive patients at 11 Italian NEN referral centers who had a histological diagnosis of well-differentiated pancreatic NETs (pNETs), no evidence or history of familiar NEN syndromes, treated with at least two consecutive therapeutic lines with approved agents, and evidence of progressive disease according to Response Evaluation Criteria in Solid Tumors (RECIST) criteria, version 1.1 [7] before change of treatment. The following baseline characteristics have been collected: sex, age at the time of the start of first-line treatment, pathology and grading, presence of symptoms secondary to hormone hypersecretion, surgical resection of primary tumor, disease stage at the time of first-line treatment start, presence of metastases outside of liver (extrahepatic disease extension), type of treatment received and outcomes (start and end date, best response, date of evidence of radiological progression of disease or last radiological evaluation), and date of death or last follow-up. The histological specimens were examined by a NEN-dedicated pathologist at each center. Tumors were classified according to the WHO 2019 classification and the ENETS grading system [1,8]. The Ki67 proliferation index was expressed as a percentage based on the count of Ki67-positive cells on 2000 tumor cells in the areas of the highest immunostaining. Clinical and radiological work-up has been performed according to the most recently available ENETS guidelines [9,10,11,12,13]. Objective response rate (ORR) was defined as the proportion of patients who achieved complete response (CR) or partial response (PR) as the best radiological response during each treatment line. Progression-free survival (PFS) was defined as the interval between the start of the therapy and the time of progression of disease (PD) or death by any cause, whichever occurred first. Overall survival (OS) was defined as the time between first-line treatment start and death by any cause. All patients or their legal representatives provided written informed consent for anonymous review of their data for research purposes. The study protocol was approved by the local Institution Review Board (Comitato Etico Indipendente, S.Orsola-Malpighi University Hospital, Bologna) and was conducted in accordance with the principles of the Declaration of Helsinki (6th revision, 2008).

### 2.2. Statistical Analysis

Categorical variables were expressed as numbers (percentage). Continuous variables were reported as median and range. Survival estimates were made using the Kaplan–Meier method, and the results were compared using the log-rank test. Predictive risk factors for PD and death were evaluated by univariate and multivariate analysis using the Cox proportional hazards method. Risk factors were expressed as hazard ratios (HRs) and 95% confidence interval (95% CI). The multivariate model was fitted using the forward stepwise method after including all variables. All analyses carried out for predictive and risk factors are listed in the tables. The *p* value was considered significant when inferior to 0.05. Statistical analysis was performed using dedicated software (IBM—SPSS Statistics v. 22). 

## 3. Results

### 3.1. Study Population

Between January 1997 and December 2020, 237 consecutive patients with pNETs matched the inclusion criteria, but 36 were excluded because they had received investigational treatments. Thus, 201 patients were included in the study, whose baseline characteristics are summarized in Table 1. One hundred eighteen patients (58.7%) were males. Median age at diagnosis was 55 years (range 20–86), while median age at start of the first line of therapy was 58 (range 21–86). As for WHO classification, 40 patients (19.9%) had a G1 NET, while 149 (74.1%) had a G2 NET; data were missing in 12 cases (6.0%). Median Ki67 was 5% (range 0.6–20%); 144 patients (71.6%) had Ki67 ≤ 10%, whereas 45 (22.4%) had Ki67 > 10%. Twenty-seven patients (13.4%) had symptoms related to hormone hypersecretion: 9 hyperinsulinemic hypoglycaemia, 6 carcinoid syndrome, 4 Cushing syndrome, 4 Zollinger–Ellison syndrome, 2 glucagone hypersecretion, and 2 VIP hypersecretion. Primary tumor was resected in 98 patients (48.8%), and 48 patients (23.9%) had extrahepatic disease extension at diagnosis. Among patients undergoing surgery of primary tumors, 58 received radical surgery (R0), while 40 underwent surgery in the context of metastatic disease. At the beginning of first-line treatment, 192 patients had stage IV disease, whereas 9 had stage IIIB disease.

### 3.2. First-Line Therapy

In the first-line setting, 128 patients (63.7%) received standard-dose SSA alone, 35 (17.4%) SSA + RLT, 21 (10.4%) temozolomide (TMZ)-based chemotherapy, and 17 (10.4%) SSA + targeted therapy (TT) (n = 16 everolimus, n = 1 sunitinib). Overall, median PFS to first-line treatment (PFS1) was 13.9 months (95% CI: 10.2–17.6), while ORR was 14.9%: 4 patients achieved CR and 26 PR. In particular, ORR was 9.1% in patients who received SSA, 37.1% in those who received RLT, 0% in those who received TT, and 31.6% in those treated with chemotherapy. A difference in PFS was observed according to grading (G1: 22.0 months [95% CI: 15.3–28.7] vs. G2: 11.0 months [95% CI: 7.6–14.4]; *p* = 0.002; Figure 1A), surgery of primary tumor (yes: 17.6 months [95% CI: 12.1–23.1] vs. no: 11.7 months [95% CI: 8.1–15.3]; *p* = 0.044; Figure 1B), Ki67 ≤ 10% (yes: 15.9 months [95% CI: 10.0–21.8] vs. no: 5.9 months [95% CI: 1.6–10.2]; Figure 1C), and type of treatment (SSA: 9.7 months [95% CI: 6.9–12.5; RLT 34.3 [95% CI: 27.3–41.2]; TT: 10.1 [95% CI: 7.3–12.9]; CHT 9.1 [95% CI: 10.2–17.6]; *p* < 0.001; Figure 1D). No significant PFS difference according to sex (*p* = 0.189), presence of syndrome (*p* = 0.651), or presence of extrahepatic metastatic disease (*p* = 0.175) was observed. A higher risk for progression was associated with Ki67 > 10% (HR 2.04; *p* < 0.001) and first-line therapy with chemotherapy compared to SSA (HR 1.70; *p* = 0.029, Table 2). A lower risk of progression was associated with resection of primary tumor (HR 0.75; *p* = 0.046) and first-line therapy with RLT compared to SSA (HR 0.54; *p* = 0.001). Sex, presence of symptoms related to hormone hypersecretion, presence of extrahepatic metastatic disease, and first-line therapy with TKI were not significantly associated with the risk of progression. After adjusting for potential confounding factors, Ki67 > 10% (HR: 1.98; *p* < 0.001), resection of primary tumor (HR: 0.69; *p* = 0.019), and first-line therapy with RLT (HR: 0.49; *p* < 0.001) retained significant associations with the risk of progression. 

### 3.3. Second-Line Therapy

Ninety-four patients (46.8%) received SSA + targeted therapy, 35 chemotherapy (either TMZ- or oxaliplatin-based) (17.4%), 45 SSA + RLT (22.4%), and 27 SSA alone-based strategy (either non-conventional-dose SSA or switched SSA; 13.4%) as second-line treatment. Overall, median PFS to second-line treatment (PFS2) was 15.0 months (95% CI: 10.7–19.3 months), while ORR was 5.5%: 1 patient achieved CR and 10 PR. Significantly longer PFS2 was associated with grading (G1: 24.1 months [95% CI: 14.4–33.8] vs. G2: 12.0 months [95% CI: 9.2–14.8]; *p* = 0.006; Figure 2A), Ki67 ≤ 10% (18.4 months [95% CI: 12.5–24.3] vs. 9.9 months [95% CI: 6.2–13.6]; *p* = 0.025; Figure 2B) and type of therapy (SSA HD/switch 10.0 months; RLT 26.0 months; TT: 16.0 months; chemotherapy 7.7 months; *p* = 0.001; Figure 2C). No significant PFS2 difference according to sex (*p* = 0.232), presence of syndrome (*p* = 0.961), surgery of primary tumor (*p* = 0.141), or presence of extrahepatic metastatic disease (*p* = 0.204) was observed. A higher risk of progression to second-line treatment was associated with Ki67 > 10% (HR 1.54; *p* = 0.027), while a lower risk of progression was associated with second-line treatment with RLT as compared to other therapeutic strategies (Table 3). Sex, presence of symptoms related to hormone hypersecretion, surgery of primary tumor, and presence of extrahepatic metastatic disease were not associated with the risk of progression to second-line treatment. At multivariate analysis, type of second-line therapy was independently associated with the risk of progression (HD-SSA vs. RLT HR: 2.06; *p* = 0.019 and chemotherapy vs. RLT HR: 2.23; *p* = 0.003; Table 3).

### 3.4. Overall Survival

Median OS was 94.7 months (95% CI: 78.8–110.6 months). To investigate the impact of treatment sequencing on OS, we grouped patients by the three most-used treatment strategies, which encompassed 82.1% of patients (n = 165/201), as follows: SSA–RLT in 35 patients (21.2%), RLT–other in 35 (21.2%) and SSA–other in 95 (57.6%). Overall, median OS in those 165 patients was 107.9 months (95% CI: 88.2–127.5 months). Significantly longer OS was associated with sex (female: 109.1 months [95% CI: 67.9–150.3] vs. male: 86.2 months [95% CI: 67.3–105.1]; *p* = 0.031), Ki67 ≤ 10% (99.9 months [95% CI: 80.6–119.1] vs. 67.9 months [95% CI: 53.2–82.4]; *p* = 0.001), surgery of primary tumor (yes: 146.1 months [95% CI: 108.3–183.8] vs. no: 69.4 months [95% CI: 58.7–80.0]; *p* < 0.001), presence of extrahepatic metastatic disease (no: 105.2 months [95% CI: 81.7–128.6] vs. yes: 69.0 months [95% CI: 41.0–97.0]; *p* = 0.012), and type of treatment (SSA–RLT: 151.1 months; RLT–other: 86.2 months; SSA–other: 96.1 months; *p* = 0.025; Figure 3). No significant OS difference according to the presence of syndrome (*p* = 0.576) was observed. A higher risk of death was associated with male sex (HR: 1.58; *p* = 0.032), Ki67 > 10% (HR 2.18; *p* = 0.001), presence of extrahepatic metastatic disease (HR: 1.77; *p* = 0.013), and therapeutic sequence (Table 4). In particular, the RLT–other (HR: 1.80; *p* = 0.041) and SSA–other (HR: 1.62; *p* = 0.023) sequences were associated with an increased risk for death when compared to SSA–RLT (Table 5). A lower risk of death was associated with surgery of the primary tumor (HR: 0.38; *p* < 0.001). At multivariate analysis, Ki67 > 10% (HR: 2.24; *p* = 0.005) and primary tumor resection (HR: 0.40; *p* = <0.001) were independently associated with the risk of death (Table 4).

### 3.5. Overall Survival in Patients with Ki67 ≤ 10%

Because the most relevant sequences in our cohort involved the use of SSA and RLT, which are mainly adopted in the treatment of pNETs with Ki67 ≤ 10%, we investigated the impact of treatment sequencing in this subgroup, which comprised 80.6% of patients (n = 133/165) whose median OS was 116.4 months (95% CI: 81.4–151.3). In this subgroup, a higher risk of death was associated with the presence of extrahepatic metastatic disease (HR: 1.77; *p* = 0.045) and therapeutic sequences with SSA–other compared to SSA–RLT (Table 6), while a lower risk of death was associated with primary tumor surgery (HR: 0.34; *p* < 0.001). At multivariate analysis, primary tumor resection (HR 0.38; *p* = 0.001) and therapeutic sequence (HR 2.26; *p* = 0.039) were independently associated with the risk of death (Table 6).

### 3.6. Overall Survival in Patients with Ki67 > 10%

Finally, we investigated the impact of treatment sequencing in the subgroup of patients with Ki67 > 10%, which comprised 19.4% of patients (n = 32/165) whose median OS was 69.0 months (95% CI: 51.2–86.8). In this subgroup, no factor was found to be correlated with an increase in the risk of death in both univariate and in multivariate analysis.

## 4. Discussion

The definition of the optimal therapeutic sequence in patients with advanced, unresectable GEP-NENs remains challenging, considering the extreme heterogeneity of NETs and the lack of solid evidence supporting the choice of the most effective treatment option as first-line or beyond. We report on prospectively-collected data from a large multicenter cohort of patients with well-differentiated pNETs who received at least two treatment lines switched upon evidence of radiological progression. RLT was associated with both prolonged PFS1 and PFS2 as compared to other therapeutic strategies, namely SSA and chemotherapy, which is consistent with the findings of a recent, multicenter, large retrospective study of 508 patients with entero-pancreatic NENs and a network meta-analysis of phase III trials of different treatment strategies in GEP-NET [14,15]. The RLT–other and the SSA–other sequences were associated with an increased risk of death when compared to SSA–RLT. This is because treatment with upfront RLT in patients who had experienced disease progression with SSA is associated with significantly improved survival outcomes compared to upfront chemotherapy or targeted therapy, as demonstrated in a recent multicenter retrospective study [14]. SSAs (i.e., octreotide and lanreotide) are widely recommended by the current guidelines as first-line therapy in patients with well-differentiated, slowly progressive, G1, and G2 GEP-NENs [16,17,18,19]. Nevertheless, sequencing treatment after disease progression on SSA has not been established. The available therapeutic options beyond first-line SSA are limited, including everolimus, sunitinib, temozolomide-based chemotherapy, and nonconventional-doses SSA and RLT with lutetium 177 (^177^Lu)-DOTATATE [5,20,21,22]. Everolimus was approved based on the RADIANT 3 study, which showed a significant improvement of PFS among progressive metastatic pNET patients treated with everolimus compared to placebo (mPFS of 11.0 months vs. 4.6 months, *p* < 0.001) [23]. Similarly, sunitinib was approved based on a phase III trial showing a statistically significant improvement in PFS in progressive metastatic pNET patients treated with sunitinib compared to placebo (mPFS of 11.4 months vs. 5.5 months, *p* < 0.001) [21]. The combination of capecitabine plus temozolomide was recently compared with single-agent temozolomide in a randomized phase II trial including patients with progressive pNETs [22]. The study showed a significant improvement of PFS for the combination compared to single-agent (mPFS of 22.7 months vs. 14.4 months, *p* < 0.02), with a confirmed radiographic response rate of 40%. Furthermore, the efficacy of RLT as second-line treatment has been investigated in the NETTER-1 trial, an open-label, randomized, phase III trial, comparing RLT to high-doses octreotide among 229 patients with advanced, well-differentiated, progressive midgut NETs [5]. The primary endpoint of PFS was reached, with an estimated PFS at 20 months of 65.2% for RLT compared to 10.8% in the control arm. The final long-term analysis of NETTER-1 has shown an 11.7-month prolongation in survival among patients receiving RLT (mOS of 48 months in the experimental arm compared to 36.3 months in the control arm), although a statistically significant benefit was not achieved, likely attenuated by crossover to RLT in over a third of patients from the control arm. Although RLT is worldwide approved for both small-intestine and pNETs, prospective randomized evidence supporting its efficacy specifically in pNET is lacking. RLT use in patients with pNET has been mostly supported by retrospective data [24], or non-randomized trials, which reported that the efficacy of RLT in pNETs was similar to that reported in NETTER-1 [25]. Preliminary results from the OCLURANDOM study, a phase II randomized trial of RLT vs. sunitinib in patients with pNETs, have recently been presented [26]. Despite being a phase II trial, in contrast to the phase III NETTER-1 study, the OCLURANDOM study represents the first prospective randomized evidence of RLT in patients with pNET. The trial enrolled 84 patients (n = 41 receiving RLT and n = 43 receiving sunitinib) and demonstrated an almost double progression-free survival rate at 12 months, the primary endpoint, in the RLT arm when compared to the sunitinib arm (80.5% vs. 42%, respectively). Although definitive data are eagerly awaited, the preliminary report further supports the use of RLT in patients with pNETs.

To date, there is no evidence supporting the superiority of RLT over chemotherapy with capecitabine plus temozolomide. However, chemotherapy should be the preferred option in patients with high tumor burden requiring rapid cytoreduction. In addition, clinical trial to investigate the best sequence are challenging, as demonstrated by the SEQTOR study, which aimed to establish the better treatment sequence between everolimus followed by chemotherapy with streptozotocin-5fluoro-uracil (STZ/5-FU) and the opposite sequence in patients with pNET [27]. Slow accrual rate made the comparison of the two treatment sequences not possible, and the trial was amended to change the progression-free survival rate to first-line treatment at 12 months (12-m PFS1) as the primary endpoint. Both arms showed similar 12-m PFS1 (*p* = 0.425). STZ-5FU assigned as the first-line treatment achieved a statistically significant higher ORR compared to everolimus (30% vs. 11%, *p* = 0.014). Based on these data, STZ/5-FU should be considered as first-line treatment, especially in patients with good performance status and grade 2, when tumor shrinkage is needed.

Notably, primary tumor resection was significantly associated with longer survival. Currently, the role of surgical treatment of metastatic NENs is still controversial, considering the heterogeneous behavior of these tumors and the lack of definitive clinical evidence. However, a retrospective analysis of data from the Surveillance, Epidemiology, and End Results (SEER) database, including 897 patients with GEP-NENs and liver metastases who underwent primary tumor resection, has shown that the surgery of the primary tumor is an independent prognostic factor for survival [28]. Another retrospective analysis of data from the National Cancer Database (NCDB) has shown that the resection of the primary tumor is associated with prolonged survival among patients with G1–G2 advanced GEP-NENs [29]. Because retrospective surgical studies are subjected to bias, prospective studies are needed to clearly define the role of the surgery of the primary tumor in this setting, which might be considered in selected patients with pNET. Despite being classified as G2 pNET, tumors with Ki67 ≤ 10% have a more indolent behavior than those with higher Ki67. Since pNETs with Ki67 ≤ 10% represented the majority of our sample, we conducted a separated analysis in this group. Interestingly, the OS of pNET patients with Ki67 ≤ 10% treated with SSA–RLT was significantly longer compared to that of patients treated with other therapeutic sequences (i.e., RLT–other, SSA–other), thus suggesting that in this more indolent pNET population upfront SSA followed by RLT upon disease progression might be the best treatment choice, as recommended by ENETS Guidelines [16]. On the other hand, earlier use of RLT in the first-line setting might be more beneficial in pNETs with Ki67 > 10%. To elucidate this topic, the randomized phase III NETTER-2 trial is currently ongoing and evaluating the role of RLT as first-line treatment of patients with newly diagnosed GEP-NET and high proliferation rate (i.e., G2 >10% and G3) compared to treatment with high-dose long-acting octreotide (NCT03972488). The preliminary results of the NETTER-2 trial were recently presented at the 2024 American Society of Clinical Oncology (ASCO) Gastrointestinal Cancers Symposium [30]. The majority of tumors originated from the pancreas (54.4%) or small intestine (29.2%), whereas G3 tumors were reported in 35% of cases. Median PFS was significantly prolonged by 14.3 months, from 8.5 months in the control arm to 22.8 months in the RLT arm (*p* < 0.0001), thus reducing the risk of disease progression or death by 72%. Moreover, the ORR was significantly improved in the RLT arm (43.0%) compared to the control arm (9.3%). However, no data regarding OS are still available. Regarding the role of nonconventional-dose SSA (either by increased administered dose (dose intensity) or shortened interval between administrations (dose density), conflicting results have been reported in NEN patients after failure of the standard SSA dose. Indeed, earlier retrospective studies [31,32] reported better outcomes associated with the use of nonconventional-dose SSA as second-line treatment compared to third- or later-line treatment, which were not clearly confirmed by the recently published phase II CLARINET FORTE study. This trial investigated the efficacy of doubled-dose lanreotide (120 mg/2 weeks) among 99 patients with advanced G1 or G2 midgut NENs or pancreatic NENs, after progressing on the standard dose (120 mg/4 weeks) [33]. The median PFS was 8.3 months in midgut NENs and 5.6 months in pancreatic NENs. Overall, the estimated probability to achieve disease control rate (mostly obtained as disease stabilization) is 45%, as reported by a recent systematic review and meta-analysis evaluating the pooled data on the efficacy of increasing SSA dose in patients with progressive GEP-NENs after standard dose SSA [32]. In general, the efficacy of targeted agents (i.e., everolimus and sunitinib) in progressive GEP-NENs is well known, based on the findings reported more than ten years ago by the regulatory phase III trials, which showed an 11–16 month advantage in terms of PFS as compared to placebo [20,21,23]. However, solid data on a direct comparison between RLT and targeted agents as second-line therapy in patients with progressive disease after SSA are currently not available. A phase III, randomized trial (COMPETE trial) addressing this question is currently ongoing by comparing the efficacy of RLT and everolimus in progressive somatostatin receptor (SSTR)-positive GEP-NETs (NCT03049189). The study has recently completed accrual and results are awaited.

## 5. Conclusions

To date, the optimal management of advanced pNETs is controversial, given the heterogeneity of these tumors and the near absence of randomized trials comparing available treatment options. In patients with pNETs who received at least two treatment lines because of radiological disease progression, RLT is associated with better PFS in both first- and second-line setting. Similarly, surgical resection of the primary tumor is associated with improved survival. In the subgroup of patients with pNETs with Ki67 <10%, the best treatment sequence is represented by first-line SSA followed by RLT upon disease progression. Thus, the best treatment sequence should include upfront RLT in patients who experienced disease progression after SSA and target therapies or chemotherapy in later lines. Defining the management and treatment sequencing of advanced pancreatic NENs could be challenging and requires discussion within a multidisciplinary team, including medical and radiation oncologists, surgeons, radiologists, and nuclear medicine specialists, at referring and high-volume centers. However, larger, prospective, phase III clinical trials are needed to define the correct timing and the optimal treatment sequence of available systemic treatments in these patients.

## Figures and Tables

**Figure 1 jcm-13-02074-f001:**
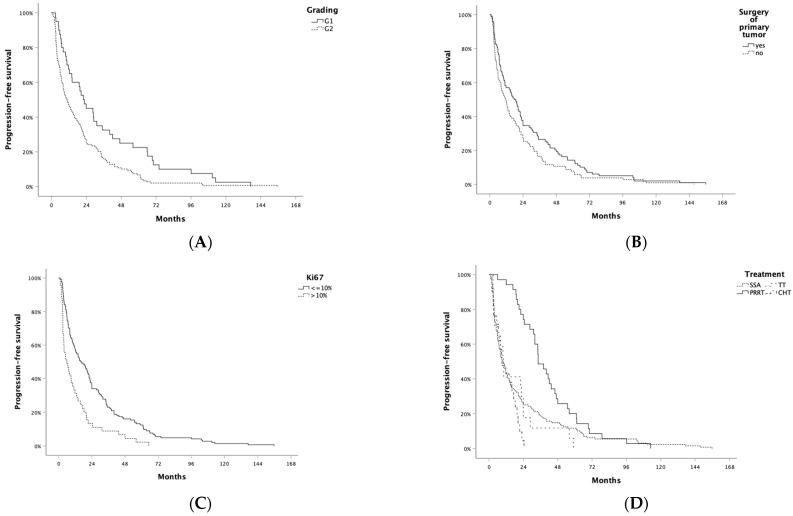
Progression-free survival to first-line treatment (PFS1) by (**A**) grading, (**B**) resection of primary tumor, (**C**) Ki67, and (**D**) type of first-line treatment. SSA: somatostatin analog; PRRT: peptide radionuclide receptor therapy; TT: targeted therapy; CHT: chemotherapy.

**Figure 2 jcm-13-02074-f002:**
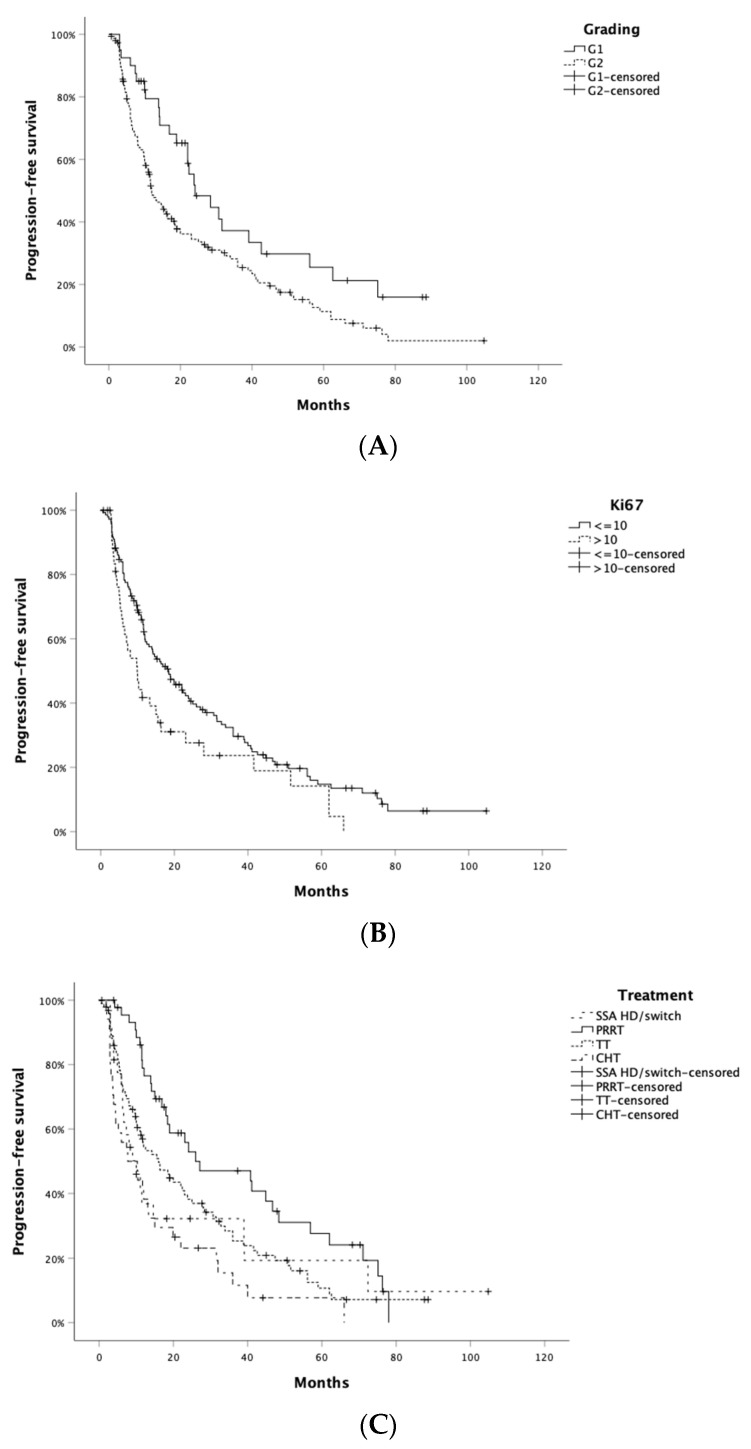
PFS to second-line treatment (PFS2) according to grade (**A**), Ki67 ≤ 10% (**B**) and type of therapy (**C**).

**Figure 3 jcm-13-02074-f003:**
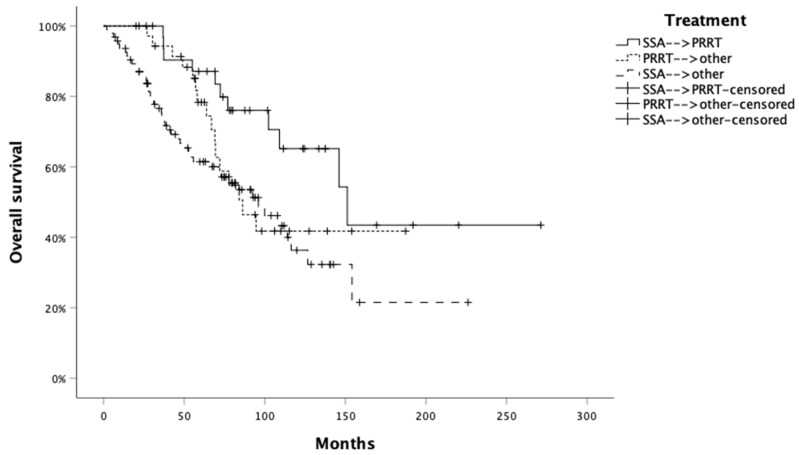
Overall survival (OS) by treatment sequence. SSA: somatostatin analog; PRRT: peptide radionuclide receptor therapy; Other includes targeted therapy, temozolomide-based or oxaliplatin-based chemotherapy, non-conventional doses SSA, or SSA switch.

**Table 1 jcm-13-02074-t001:** Patient characteristics.

Characteristic	Patients (no. 201)
Demographic:	
Gender (male), no. (%)	118 (58.7%)
Median age (range) at therapy start, years	58 (21–86)
WHO classification:	
G1, no. (%)	40 (19.9%)
G2, no. (%)	149 (74.1%)
Missing data, no. (%)	12 (6.0%)
Median Ki67%	5 (0.6–20)
Ki67 > 10%	45 (22.4%)
Functioning tumors, no. (%):	27 (13.4%)
Primary tumor surgery, no. (%)	98 (48.8%)
Extrahepatic disease, no. (%) ttggd	48 (23.9%)

**Table 2 jcm-13-02074-t002:** Predictive risk factors for disease progression during first-line therapy.

Characteristic	Univariate Analysis			Multivariate Analysis		
	HR	95% CI	*p* Value	HR	95% CI	*p* Value
Gender (male)	1.21	0.91–1.60	0.191	-	-	-
Non-functioning tumor	1.10	0.73–1.65	0.653	-	-	-
Ki67 > 10%	2.04	1.45–2.88	<0.001	1.98	1.36–2.87	<0.001
Primary tumor resection	0.75	0.57–0.99	0.046	0.69	0.51–0.94	0.019
Extrahepatic disease	1.25	0.90–1.74	0.178	-	-	-
Type of therapy						
SSA	1			1		
RLT	0.54	0.37–0.79	0.001	0.49	0.33–0.72	<0.001
TT	1.20	0.72–2.01	0.479	1.12	0.67–1.88	0.662
CHT	1.70	1.06–2.74	0.029	1.15	0.67–1.99	0.608

Abbreviations: HR: hazard ratio; 95% CI: 95% confidence interval; SSA: somatostatin analogs; RLT: radioligand therapy; TT: targeted therapy; CHT: chemotherapy.

**Table 3 jcm-13-02074-t003:** Predictive risk factors for disease progression during second-line therapy.

Characteristic	Univariate Analysis			Multivariate Analysis		
	HR	95% CI	*p* Value	HR	95% CI	*p* Value
Gender (male)	1.21	0.88–1.68	0.234	-	-	-
Non-functioning tumor	1.01	0.64–1.59	0.961	-	-	-
Ki67 > 10%	1.54	1.05–2.26	0.027	1.42	0.96–2.12	0.081
Primary tumor resection	0.79	0.57–1.08	0.143	-	-	-
Extrahepatic disease	1.26	0.88–1.82	0.207	-	-	-
Type of therapy						
RLT	1			1		
SSA HD/Switch	1.80	1.02–3.15	0.041	2.06	1.13–3.75	0.019
TT	1.62	1.07–2.47	0.023	1.48	0.96–2.23	0.074
CHT	2.67	1.62–4.42	<0.001	2.23	1.31–3.80	0.003

Abbreviations: HR: hazard ratio; 95% CI: 95% confidence interval; SSA: somatostatin analogs; HD: high dose; RLT: radioligand therapy; TT: targeted therapy; CHT: chemotherapy.

**Table 4 jcm-13-02074-t004:** Predictive risk factors for death.

Characteristic	Univariate Analysis			Multivariate Analysis		
	HR	95% CI	*p* Value	HR	95% CI	*p* Value
Gender (male)	1.58	1.04–2.04	0.032	-	-	ns
Non-functioning tumor	1.18	0.66–2.12	0.576	-	-	-
Ki67 > 10%	2.18	1.35–3.52	0.001	2.24	1.28–3.94	0.005
Primary tumor resection	0.38	0.25–0.57	<0.001	0.40	0.24–0.66	<0.001
Extrahepatic disease	1.77	1.13–2.78	0.013	-	-	-
Type of therapy						
SSA–RLT	1			1		
RLT–Other	1.80	1.02–3.15	0.041	-	-	ns
SSA–Other	1.62	1.07–2.47	0.023	-	-	ns

Abbreviations: HR: hazard ratio; 95% CI: 95% confidence interval; SSA: somatostatin analogs; RLT: radioligand therapy; ns: not significant.

**Table 5 jcm-13-02074-t005:** Outcomes according to treatment sequences.

	N. (%)	PFS Months (95% CI)	OS Months (95% CI)
SSA–RLT	35 (21.2)	62.5 (42.2–82.7)	151.1 (97.4–204.8)
RLT–Other	35 (21.2)	38.1 (28.4–47.8)	86.2 (60.8–111.7)
SSA–Other	95 (57.6)	31.2 (17.0–45.3)	96.1 (66.4–125.8)

Abbreviations: PFS: progression-free survival; OS: overall survival; 95% CI: 95% confidence interval; SSA: somatostatin analogs; RLT: radioligand therapy.

**Table 6 jcm-13-02074-t006:** Predictive risk factors for death in patients with Ki67 ≤ 10%.

Characteristic	Univariate Analysis			Multivariate Analysis		
	HR	95% CI	*p* Value	HR	95% CI	*p* Value
Gender (male)	1.32	0.80–2.20	0.280	-	-	ns
Non-functioning tumor	1.08	0.56–2.07	0.817	-	-	-
Ki67	1.04	0.96–1.13	0.371	-	-	ns
Primary tumor resection	0.34	0.20–0.57	<0.001	0.38	0.21–0.69	0.001
Extrahepatic disease	1.77	1.01–3.10	0.045	-	-	ns
Type of therapy						
SSA–RLT	1			1		
RLT–Other	2.02	0.85–4.77	0.110	1.65	0.69–3.97	0.264
SSA–Other	2.39	1.18–5.12	0.025	2.26	1.04–4.88	0.039

Abbreviations: HR: hazard ratio; 95% CI: 95% confidence interval; SSA: somatostatin analogs; RLT: radioligand therapy; ns: not significant.

## Data Availability

The data presented in this study are available on request from the corresponding author due to ethical reasons.

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
