# Peer review of "Sequencing Treatments in Patients with Advanced Well-Differentiated Pancreatic Neuroendocrine Tumor (pNET): Results from a Large Multicenter Italian Cohort"

_jcm, 2024, doi:10.3390/jcm13072074_

Round 1

Reviewer 1 Report

Comments and Suggestions for Authors

The study is well done. The topic is very interesting. 

I have some points that I want you to modify:

1. Material and methods: it must be rewrite because of the  similarities

2. I would like to see a comparison between the italian centers and other centers from Europe or outside it. 

3. Discussions are very little in debate and I would like you to write more and compare more with the literature and other studies.

4. The bibliography must be improved (more citations and more updated are needed)

Comments on the Quality of English Language

Some phrases are very log and they loose their meaning.

Some misspelling

Please review the english from the entire article.

Author Response

We thank the reviewers for their comments since we feel that thanks to them the overall quality of the manuscript has been sensibly improved.

1. Material and methods: it must be rewrite because of the similarities.

We thank the reviewer for this suggestion. We have revised the "Materials and Methods" section to improve the clarity and conciseness of the section. See lines 112-140.

2. I would like to see a comparison between the italian centers and other centers from Europe or outside it. 

We appreciate the reviewer's suggestion of comparing our Italian data with data from other European and/or global sources. While we find this suggestion highly valuable, it was not feasible within the scope of this specific study. Due to the current lack of readily available data from other countries, a direct comparison with European or worldwide data was unfortunately not possible. In addition, we were unable to identify any existing studies in the literature that would allow for such a comparison.

3. Discussions are very little in debate and I would like you to write more and compare more with the literature and other studies.

We thank the reviewer for this suggestion. We modified discussion as suggested.

4. The bibliography must be improved (more citations and more updated are needed)

We thank the reviewer for this comment. We modified bibliography accordingly.

5. Some phrases are very log and they loose their meaning. Some misspelling. Please review the english from the entire article.

We thank the reviewer for this suggestion. We improved English in the entire article.

Reviewer 2 Report

Comments and Suggestions for Authors

Thank you to the authors for conducting this interesting retrospective analysis. A few comments and questions: 

1. In the introduction, it is mentioned that the prevalence of neuroendocrine tumors is on the rise. However, this statement should also have the caveat that this in likely in part due to detection of these neoplasms has also increased (i.e. incidental findings on sensitive imaging)

2. Under study design (2.1) and results (3.1), what is meant by "all consecutive patients" at the 11 included centers?

3. "Primary tumor surgery" should be described/defined - for example, does this included enucleation? If possible, would be interesting to report the break down of types of primary tumor surgery (i.e. enucleation, distal pancreatectomy etc)

4. It is unclear why the RLT-other or SSA-other groups had increased risk for death compared to SSA-RLT group, could the authors explain this finding. 

5.  In Figure 3 - Fig3A-C are unsurprising and are all characteristics that have been previously established, though I think important to have as reference (i.e. in Table 4). Do these Kaplan-Meier curves need to be included since it is in Table 4, Fig 3D should be highlighted. 

Comments on the Quality of English Language

Minor grammatical errors can be addressed in this manuscript. 

Author Response

1. In the introduction, it is mentioned that the prevalence of neuroendocrine tumors is on the rise. However, this statement should also have the caveat that this in likely in part due to detection of these neoplasms has also increased (i.e. incidental findings on sensitive imaging)

 We thank the reviewer for this suggestion. We modified introduction as suggested.

2. Under study design (2.1) and results (3.1), what is meant by "all consecutive patients" at the 11 included centers?

We thank the reviewer for this comment. “All consecutive patients” means that they met the inclusion criteria and signed the informed consent. 

3. "Primary tumor surgery" should be described/defined - for example, does this included enucleation? If possible, would be interesting to report the break down of types of primary tumor surgery (i.e. enucleation, distal pancreatectomy etc.

We thank the reviewer for this comment. Unfortunately, this data is not available but we have only data about surgery of the primary tumor.

4. It is unclear why the RLT-other or SSA-other groups had increased risk for death compared to SSA-RLT group, could the authors explain this finding. 

We thank the reviewer for this comment. The SSA-RLT sequence is associated with a lower risk for death compared to other sequences because it is associated with a longer overall survival (Lines 525-529).

5. In Figure 3 - Fig3A-C are unsurprising and are all characteristics that have been previously established, though I think important to have as reference (i.e. in Table 4). Do these Kaplan-Meier curves need to be included since it is in Table 4, Fig 3D should be highlighted. 

We thank the reviewer for this suggestion. We have modified the manuscript as suggested, highlighting Figure 3D.

Reviewer 3 Report

Comments and Suggestions for Authors

Thank you for the opportunity to review the article entiltled “Sequencing treatments in patients with advanced well-differentiated pancreatic neuroendocrine tumor (pNET): results from 3 a large multicenter Italian cohort”.

The authors conducted a retrospective multi-center study regarding the Sequencing of the treatments in patients with advanced well-differentiated pancreatic neuroendocrine tumor (pNET). The identification of the best treatment timing and sequence in patients with pNET  is still unknown and we greatly need data showing what the optimal line of treatment should be like.

I just have a few minor questions:

-          do all patients had the reassessment of pathology according to the 2019 WHO classification?

-          what was the real number of patients with baseline disseminated pNET I mean still disseminated disease after surgery, not just metastases found in surgical specimens – could you provide the information about disease stage at the start of the first-line treatment

-          did the removal of the primary lesion apply to patients with dissemination in all cases?

-          could you compare in a table the effectiveness (in terms of PFS and OS) of the 3 most used treatment strategies of first and second line treatment

-          could you provide data (as in chapter 3.5) on patients with Ki67>10?

Author Response

We thank the reviewers for their comments since we feel that thanks to them the overall quality of the manuscript has been sensibly improved.

1. Do all patients had the reassessment of pathology according to the 2019 WHO classification?

We thank the reviewer for this comment. Yes, all patients had the reassessment of pathology according to the 2019 WHO classification (line 122)

2. What was the real number of patients with baseline disseminated pNET I mean still disseminated disease after surgery, not just metastases found in surgical specimens – could you provide the information about disease stage at the start of the first-line treatment.

We thank the reviewer for this comment. Among the examined patients, 58 received radical surgery (R0), 40 underwent surgery of the primary lesion in the presence of metastatic disease. At the beginning of first-line treatment, 192 patients had stage 4 disease, and 9 had stage IIIB disease. We have modified text accordingly (Lines 168-171).

2. Did the removal of the primary lesion apply to patients with dissemination in all cases?

We thank the reviewer for this comment. Out of the total 201 patients, 98 underwent surgery for the primary lesion. Among these, surgery resulted in R0 status in 58 patients, while residual disease was observed in 40 of cases after surgery.

4. Could you compare in a table the effectiveness (in terms of PFS and OS) of the 3 most used treatment strategies of first and second line treatment

We thank the reviewer for this suggestion. We created table 5 to compare the efficacy in terms of OS and PFS of the most commonly used treatments.

5. could you provide data (as in chapter 3.5) on patients with Ki67>10?

We thank the reviewer for this suggestion. We added the chapter 3.6, dedicated to OS in patients with Ki-67 > 10%.

Round 2

Reviewer 1 Report

Comments and Suggestions for Authors

The authors made the required corrections.